# On Smoothing and Habit Formation of Variable Life Annuity Benefits

**Mogens Steffensen †** and **Savannah Halling Vikkelsøe *,†**

Department of Mathematical Sciences, University of Copenhagen, 2100 Copenhagen Ø, Denmark;
mogens@math.ku.dk
* Correspondence: xgl102@alumni.ku.dk
† These authors contributed equally to this work.

**Abstract:** This paper studies optimal consumption and investment strategies with lifetime uncertainty to design a smooth pension product. In a simplified Black–Scholes market, we investigate three strategies for consumption and investment: the classical strategy, the habit strategy, and the hybrid strategy. Incorporating additive habit formation in preferences leads to a request for less consumption volatility. Studying the consumption dynamics, it turns out that the hybrid strategy complies with the same preferences as the habit strategy. In our design of a smooth pension product, we are highly inspired by the consumption structure under the hybrid strategy and let consumption be specified as a time-dependent weighted average of last year's consumption level and a standard market rate life annuity. We give two approaches for the investment portfolio. The numerical examples show that consumption under these approaches is less volatile than consumption under the classical strategy.

**Keywords:** optimal consumption and investment; consumption dynamics; smooth pension product; decumulation phase

## 1. Introduction

The Danish Financial Supervisory Authority frequently publishes an evaluation of the current financial risks that dominate the Danish pension and insurance market. In their latest evaluation, DFSA (2022a), the focus is on the interest rate, inflation, and the risk of recession. These factors create a volatile and insecure financial market. Thus, Danish pension companies are encouraged to revisit the pension products they offer and reassess whether these meet the investor's potential preferences for smoothing. The aim is to maintain the investor's interests and keep the investor's faith in the pension companies.

Most Danish pension companies offer market-rate pension products, where the size of the pension benefit depends on the return on investing pension savings in risky assets. Such products are comparable with variable life annuities. Thus, increased volatility and insecurity in the financial market might increase the need for the stability of pension benefits. In terms of optimal stochastic control theory with lifetime uncertainty, we think of the pension benefits as the consumption of an investor.

With stochastic optimal control theory, this paper aims to present optimal consumption and investment strategies that inspire the design of a smooth pension product. Merton (1969, 1971) formalized and solved the original problem of maximizing the total expected utility of intertemporal consumption and terminal wealth of an investor over a fixed time interval. Lifetime uncertainty was introduced to the stochastic optimal control problem by Richard (1975). The result of optimal consumption from the above literature resembles the design of a standard market rate life annuity offered by most Danish pension companies.

Introducing additive habit formation in preferences to the stochastic optimal control problem can explain the request for smooth consumption; see Bruhn and Steffensen (2013).

The habit level of an investor depends on past consumption. By introducing habit formation to the problem, we assume that the investor only obtains utility from the current consumption level over the current habit level. Thus, we consider the current consumption level relative to past consumption. Munk (2008) studies optimal consumption and investment strategies with additive habit formation in preferences and stochastic variations in investment opportunities. He concludes that habit formation in preferences reduces the speculative investments of the investor to ensure the habit level as a minimum level of consumption.

Kashif et al. (2020) investigate different optimal consumption and investment strategies for university endowment funds. Their objective is to find a strategy that smooths consumption over time. They present an ad hoc consumption strategy called the hybrid strategy. Consumption is specified as a weighted average of last year's consumption level and a constant proportion of the current wealth of the investor. They show that the hybrid strategy is the least volatile strategy for consumption compared to the optimal consumption strategy in Merton (1969, 1971) and to consumption defined as a constant proportion of the current wealth. These strategies can be translated to pension product designs by introducing lifetime uncertainty.

This paper aims to understand which preferences lead to the request for consumption stability. We can only design a smooth pension product that meets these preferences with such an understanding. A main contribution is to connect the hybrid strategy suggested by Kashif et al. (2020) with the consumption profile arising from habit formation in the preferences. We realize through studying the consumption dynamics that the products they form overlap. This suggests continuing the design work within the class of hybrid strategies since we then know more about which preferences they satisfy.

We aim for a smooth pension product that is feasible, transparent, and fair from the perspective of both the investor and the pension company. An already existing smooth pension product is Tidspension, as discussed by Bruhn and Steffensen (2013). This pension product includes a buffer account to achieve a smoothing effect on consumption, increasing the design's complexity. We seek for even more simplicity.

The rest of the paper is structured as follows. Section 2 sets up the general framework of the investment market and the utility maximization problem and introduces lifetime uncertainty to the problem. Section 3 presents three strategies for consumption and investment: the classical strategy, the habit strategy, and the hybrid strategy. Continuing to Section 4, the dynamics of consumption under the three strategies are studied. Moreover, the similarities between the consumption dynamics under the habit and hybrid strategies are investigated. In Section 5, the three strategies inspire two approaches for a smooth pension product. Section 6 presents numerical examples of the classical strategy and the two approaches. The conclusion of the results is given in Section 7. Appendix A contains the proofs of the consumption dynamics from Section 4.

## 2. The Model

Initially, we set up the general framework of the investment market and the utility maximization problem. Moreover, we introduce lifetime uncertainty to translate the results into designs of pension products.

### 2.1. The General Framework

This paper considers an investor who invests continuously in a complete Black–Scholes market over a fixed time interval $[0, T]$. The risk-free asset $B_t$ (a bond) follows the dynamics given by

$$dB_t = rB_t dt,$$
$$B_0 = b_0.$$

The risky asset $S_t$ (a stock) is governed by

$$dS_t = \lambda S_t dt + \sigma S_t dW_t,$$
$$S_0 = s_0,$$

where $W_t$ is a Brownian motion under the probability space $(\Omega, \mathcal{F}, P)$. For simplicity, we assume that the interest rate $r$, the mean rate of return $\lambda$, and the volatility $\sigma$ are constants. We consider this rather simple investment market to avoid unnecessary complexity of the calculations and the associated results. See, e.g., Munk (2008) for a similar utility maximization problem in a more complex investment market with higher dimensions of the risky asset and stochastic interest rate.

The investor has initial wealth $x_0$ at time $t = 0$. The wealth of the investor at time $t$ is denoted by $X_t$, and the proportion of wealth invested in the risky asset at time $t$ is denoted by $\pi_t$. The remaining proportion, $1 - \pi_t$, is invested in the risk-free asset. The investor has consumption rate $c_t$ at time $t$. Hence, wealth evolves by

$$dX_t = (r + \pi_t(\lambda - r))X_t dt + \pi_t \sigma X_t dW_t - c_t dt,$$
$$X_0 = x_0. \tag{1}$$

The investor allocates consumption and investment over $[0, T]$ to maximize the expected utility. Merton (1969, 1971) formalized this utility maximization problem by

$$\max_{c, \pi} E\left[ \int_0^T U(t, c_t) dt + L(X_T) \right], \tag{2}$$

subject to the wealth dynamics (1) and the constraint $c_t \geq 0$ for all $t \geq 0$. Here, $U$ and $L$ are interpreted as the utility functions for consumption and terminal wealth, respectively. The problem given by (2) is a so-called stochastic optimal control problem, and the dynamic programming technique solves it.

Throughout the paper, we consider power utility, which is characterized by a constant relative risk aversion $\gamma$ (CRRA), constant elasticity of intertemporal substitution $\gamma^{-1}$ (EIS), and a constant subjective discount rate of time preference $\rho \geq 0$, i.e., the impatience factor. Hence, the utility functions for consumption and terminal wealth are specified as

$$U(t, c) = e^{-\rho t} \frac{1}{1 - \gamma} c^{1-\gamma} \quad \text{and} \quad L(x) = e^{-\rho T} \frac{1}{1 - \gamma} x^{1-\gamma}, \tag{3}$$

with $\gamma \in (0, \infty) \setminus \{1\}$. For $\gamma = 1$, we have logarithmic utility, since $\lim_{\gamma \to 1} \frac{1}{1-\gamma}(c^{1-\gamma} - 1) = \log(c)$. In Section 3, we derive the optimal controls for $\gamma \in (0, \infty) \setminus \{1\}$, but we observe that these also hold for $\gamma = 1$.

To derive the solution of (2), the problem is embedded in an (optimal) value function given by

$$V(t, x) = \sup_{c, \pi} E_{t,x}\left[ \int_t^T e^{-\rho(s-t)} \left( \frac{1}{1-\gamma} c_s^{1-\gamma} ds + \frac{1}{1-\gamma} X_s^{1-\gamma} d\varepsilon_T(s) \right) \right], \tag{4}$$

where $\varepsilon_T(\cdot) = 1_{\{T \leq \cdot\}}$ and $E_{t,x}$ is the conditional expectation given that $X_t = x$. Regarding the integral of the terminal wealth, we have used the same notation as Bruhn and Steffensen (2013). The value function is the indirect utility from wealth at $t \in [0, T]$. Hence, the value function is often called the indirect utility function in the literature.

## 2.2. Introducing Lifetime Uncertainty

To consider the above framework in life insurance, we introduce lifetime uncertainty to the utility maximization problem. Let the investor's lifetime be denoted by the non-

negative random variable $\tau$ defined on the probability space $(\Omega, \mathcal{F}, P)$. The mortality rate or the hazard function is then defined by

$$\mu_t = \lim_{z \to 0} \frac{P(t \le \tau < t + z \mid \tau \ge t)}{z}. \tag{5}$$

As the investor has no utility from bequest, we add the income of $\mu_t^* X_t$ to the wealth dynamics (1) (see, e.g., Konicz et al. 2015). The intuition is that if there is no utility from bequest, the investor is willing to give up the wealth in case of death. For this wealth, the insurance company is willing to pay the so-called mortality credits. They work as a premium payment for reversed life insurance. It can be shown that $X_t$ is the optimal life insurance sum (see, e.g., Richard (1975) or Kraft and Steffensen (2008)). The function $\mu_t^*$ is called the pricing intensity and is decided by the insurance company. Then, the wealth evolves by

$$dX_t = (r + \mu_t^* + \pi_t(\lambda - r))X_t dt + \pi_t \sigma X_t dW_t - c_t dt,$$
$$X_0 = x_0. \tag{6}$$

Now, given the modified wealth dynamics (6), the value function is given by

$$V(t, x) = \sup_{c, \pi} E_{t,x} \left[ \int_t^T e^{-\int_t^s (\rho + \mu_u) du} \left( \frac{1}{1 - \gamma} c_s^{1-\gamma} ds + \frac{1}{1 - \gamma} X_s^{1-\gamma} d\varepsilon_T(s) \right) \right], \tag{7}$$

where $E_{t,x}$ is the conditional expectation given that the investor is alive at time $t \in [0, T]$ and holds wealth $X_t = x$. Here, we assume that the investor retires at time 0. Thus, we focus on the decumulation phase of the investor. The utility functions are multiplied by the conditional probability that the investor survives from time $t$ to $s$ given that she is alive at time $t \le s$, i.e., $\exp(-\int_t^s \mu_u du)$. Reasonably, the investor only obtains utility from consumption and terminal wealth as long as she is alive.

This paper aims to solve the utility maximization problem given by (7) under three different strategies and interpret the optimal controls as the pension benefits and the investment portfolio in the decumulation phase of the investor. We consider the classical strategy derived directly from (7) without terminal wealth in contrast to two strategies carefully chosen to achieve a smoothing effect on consumption.

## 3. Three Strategies for Consumption and Investment

In this section, we solve the utility maximization problem given by (7) under three different strategies. The first strategy is derived directly from the situation without terminal wealth and is considered classical. In the second strategy, we allow for additive habit formation in preferences. We will refer to this strategy as the habit strategy. Continuing to the third strategy, we examine the hybrid strategy, which is an ad hoc smoothing strategy for consumption presented by Kashif et al. (2020).

### 3.1. The Classical Strategy

Initially, we consider the classical strategy and solve the value function given by

$$V(t, x) = \sup_{c, \pi} E_{t,x} \left[ \int_t^T e^{-\int_t^s (\rho + \mu_u) du} \frac{1}{1 - \gamma} c_s^{1-\gamma} ds \right], \tag{8}$$

subject to the wealth dynamics (6) and the constraint $c_t \ge 0$ for all $t \ge 0$. Here, we assume that the investor has utility solely from intertemporal consumption, i.e., we omit utility

from terminal wealth. Using the technique of dynamic programming, we find and solve the corresponding Hamilton–Jacobi–Bellman (HJB) equation given by

$$0 = V_t - (\rho + \mu)V + \sup_{c,\pi} \left\{ \frac{1}{1-\gamma} c^{1-\gamma} + ((r + \mu^* + \pi(\lambda - r))x - c)V_x \right.$$
$$\left. + \frac{1}{2}\pi^2 \sigma^2 x^2 V_{xx} \right\}, \tag{9}$$
$$V(T, x) = 0.$$

The dependencies have been suppressed for ease of notation, and the subscripts denote the partial derivatives. The HJB equation agrees with the one in, e.g., Merton (1969), Kashif et al. (2020) with $\mu_t = 0$ and $\mu_t^* = 0$ for all $t \geq 0$. The structure is completely maintained when we introduce lifetime uncertainty. It can also be found in Steffensen and Søe (2023) with the mortality effect present. Therefore, we neither establish a verification proof nor derive the solution to the HJB equation. We simply state its solution and the corresponding optimal controls in the following. The solution to the HJB Equation (9) is given by the following value function:

$$V(t, x) = \frac{1}{1-\gamma} a_1(t)^\gamma x^{1-\gamma}, \tag{10}$$

where

$$a_1(t) = \int_t^T e^{-\int_t^s (\tilde{r} + \tilde{\mu}_u) du} ds, \tag{11}$$

with

$$\tilde{r} := \frac{1}{\gamma}\rho + \left(1 - \frac{1}{\gamma}\right)\left(r + \frac{1}{2}\frac{\theta^2}{\gamma}\right), \tag{12}$$

$$\tilde{\mu}_t := \frac{1}{\gamma}\mu_t + \left(1 - \frac{1}{\gamma}\right)\mu_t^*. \tag{13}$$

Here, we have defined $\theta := (\lambda - r)/\sigma$, which is the market price of risk. The optimal consumption and investment portfolio, i.e., the optimal controls, under the classical strategy are given by

$$c_t^* = \frac{X_t}{a_1(t)}, \tag{14}$$

$$\pi_t^* = \frac{\lambda - r}{\sigma^2 \gamma}. \tag{15}$$

We emphasize that the value function and the optimal controls correspond to the results in, e.g., Merton (1969) and Kashif et al. (2020) with $\mu_t = 0$ and $\mu_t^* = 0$ for all $t \geq 0$, as well as with Steffensen and Søe (2023).

Observe that optimal consumption at time $t$ equals the wealth at time $t$. The proportion is determined by the function $a_1$, which has the interpretation of an annuity with the utility-adjusted interest rate $\tilde{r}$ as the calculation rate and the utility-adjusted mortality $\tilde{\mu}_t$ as the calculation of mortality. The utility-adjusted interest rate is a weighted average of two interest rate factors. These are the impatience factor and an interest rate obtained from investing in the financial market. The utility-adjusted mortality rate is a weighted average of the original mortality rate and the pricing intensity. The weight is determined by $\gamma^{-1}$. Optimal consumption under the classical strategy coincides with a standard market rate life annuity structure. The classical strategy considers optimal expected utility.

Moreover, the optimal proportion of wealth invested in the risky asset is constant under the classical strategy. We refer to this constant as Merton's constant. See Merton (1969, 1971).

### 3.2. The Habit Strategy

Now, we introduce the habit formation of the investor to the utility maximization problem. Incorporating habit formation is an additive form with respect to consumption to achieve a smoothing effect on the optimal consumption. We refer to the strategy with additive habit formation in preferences as the habit strategy. As Munk (2008), we define the habit level by

$$h_t = h_0 e^{-\beta t} + \alpha \int_0^t e^{-\beta(t-s)} c_s ds, \tag{16}$$

where $h_0$, $\alpha$ and $\beta$ are assumed to be non-negative constants. Observe that the habit level at time $t$ is the sum of the discounted, initial habit level $h_0$ and the discounted, weighted consumption rates from time 0 to $t$. The constant $\alpha$ is interpreted as the weight providing the relative importance to past consumption, while $\beta$ is the discount rate describing the decline in the relative importance to past consumption. Hence, $\alpha$ and $\beta$ are parameters measuring the habit preferences of the investor. The habit level follows the dynamics given by

$$dh_t = (\alpha c_t - \beta h_t)dt, \tag{17}$$
$$h_0 > 0.$$

We include the habit level as an additional state variable in the utility maximization problem. Then, the value function becomes

$$V(t, x, h) = \sup_{c, \pi} E_{t,x,h} \left[ \int_t^T e^{-\int_t^s (\rho + \mu_u) du} \frac{1}{1-\gamma} (c_s - h_s)^{1-\gamma} ds \right],$$

subject to the wealth dynamics (6), the habit dynamics (17) and the constraint $c_t \geq 0$ for all $t \geq 0$. Here, $E_{t,x,h}$ is the conditional expectation given that the investor is alive at time $t$, holds wealth $X_t = x$, and has habit level $h_t = h$. Again, exploiting the technique of dynamic programming, we find and solve the corresponding HJB equation given by

$$0 = V_t - (\rho + \mu)V + \sup_{c, \pi} \left\{ \frac{1}{1-\gamma}(c-h)^{1-\gamma} + ((r + \mu^* + \pi(\lambda - r))x - c)V_x \right.$$
$$\left. + (\alpha c - \beta h)V_h + \frac{1}{2}\pi^2\sigma^2 x^2 V_{xx} \right\}, \tag{18}$$
$$V(T, x) = 0.$$

The HJB equation agrees with the one in, e.g., Munk (2008) and Kashif et al. (2020) with $\mu_t = 0$ and $\mu_t^* = 0$ for all $t \geq 0$. The structure is completely maintained when we introduce lifetime uncertainty. Therefore, we neither establish a verification proof nor derive the solution to the HJB equation. We simply state its solution and the corresponding optimal controls in the following. The solution to the HJB Equation (18) is given by the following value function:

$$V(t, x, h) = \frac{1}{1-\gamma} a_2(t)^\gamma (x - hb_2(t))^{1-\gamma},$$

where

$$a_2(t) = \int_t^T e^{-\int_t^s (\tilde{r} + \tilde{\mu}_u) du} (1 + \alpha b_2(s))^{1 - \frac{1}{\gamma}} ds, \tag{19}$$

$$b_2(t) = \int_t^T e^{-\int_t^s (r + \mu_u^* + \beta - \alpha) du} ds, \tag{20}$$

with $\tilde{r}$ and $\tilde{\mu}$ defined as in (12) and (13), respectively. The optimal consumption and investment portfolio under the habit strategy are given by

$$c_t^* = h_t + (1 + \alpha b_2(t))^{-\frac{1}{\gamma}} \frac{X_t - h_t b_2(t)}{a_2(t)}, \tag{21}$$

$$\pi_t^* = \frac{\lambda - r}{\sigma^2 \gamma} \frac{X_t - h_t b_2(t)}{X_t}. \tag{22}$$

We emphasize that the value function as well as the optimal controls are directly comparable with the ones obtained by, e.g., Munk (2008) and Kashif et al. (2020) with $\mu_t = 0$ and $\mu_t^* = 0$ for all $t \geq 0$.

The optimal consumption at time $t$ under the habit strategy is to consume at the current minimum level, i.e., the habit level at time $t$, plus a proportion of the current wealth over the price of maintaining future minimum consumption, i.e., $X_t - h_t b_2(t)$. The proportion is determined by the fraction $(1 + \alpha b_2(t))^{-1/\gamma}/a_2(t)$. We still interpret the function $a_2$ as an annuity with the utility-adjusted interest rate $\tilde{r}$ as the calculation rate and the utility-adjusted mortality rate $\tilde{\mu}_t$ as the calculation of mortality. Still, an extra term is added to the integral due to incorporating additive habit formation. The function $b_2$ is an annuity determined by the habit preferences of the investor, the interest rate, and the pricing intensity. Thus, multiplying the current habit level by $b_2$ is interpreted as the discounted, current habit level, i.e., the price of maintaining future minimum consumption as mentioned above. Discounting depends on the investor's habit preferences. Moreover, observe that optimal consumption under the habit strategy is affine in wealth. This implies a smoothing effect on consumption (see Bruhn and Steffensen 2013).

Unlike the classical strategy, optimal investment under the habit strategy is time-dependent. Both strategies have Merton's constant in common, but only $X_t - h_t b_2(t)$ is invested in the risky asset as the habit level is added to the strategy. Thus, habit formation reduces risk tolerance to secure consumption at the habit level as a minimum.

### 3.3. The Hybrid Strategy

This section presents an ad hoc smoothing strategy for consumption introduced by Kashif et al. (2020), who study different optimal consumption and investment strategies for university endowment funds. First, consider their fixed consumption–wealth ratio strategy with consumption rate specified as a proportion of the wealth, i.e.,

$$c_t = y X_t, \tag{23}$$

where $y$ is a constant determining the consumption–wealth ratio. Then, Kashif et al. (2020) extend this strategy by a simplified version of the Yale/Stanford rule from Cejnek et al. (2014) and define the discrete-time consumption rate by

$$c_t = w c_{t-1} + (1 - w) y X_t, \tag{24}$$

where $w$ is a weight. Thus, consumption is a weighted sum of the consumption in the previous year and a fixed proportion of the wealth in the current year. Similarly to Kashif et al. (2020), we will refer to this ad hoc strategy for consumption as the hybrid strategy, since it is a hybrid between last year's consumption rate and the fixed consumption–wealth ratio strategy.

Observe that $w$ and $y$ are considered to be constants. In later sections, we will allow $w$ and $y$ to be time-dependent such that (24) is more flexible in designing a pension product. For now, we keep $w$ and $y$ constants.

As consumption is already predefined by (24) before solving the utility maximization problem, we include consumption as an additional state variable, such as wealth. Thus, we assume that (24) is the optimal way to consume, and we find the corresponding optimal way to invest by maximizing the expected utility of terminal wealth with both the wealth and consumption as state variables (see Kashif et al. 2020). We must find the consumption dynamics to include consumption as a state variable. By Itô's formula, the dynamics of $c$ are given by

$$dc_t = \frac{\partial}{\partial t}c_t dt + \frac{\partial}{\partial x}c_t dX_t + \frac{1}{2}\frac{\partial^2}{\partial x^2}c_t(dX_t)^2.$$

We can calculate the partial derivatives of $c$ by considering (24). For the time-derivative, we consider the discrete-time version

$$c_t = (1 - (1-w)\Delta)c_{t-\Delta} + (1-w)\Delta y X_t,$$

where we scale the weight on current wealth with the length of the timestep. Note that (24) is obtained as $\Delta = 1$. Now, we can calculate

$$\frac{c_t - c_{t-\Delta}}{\Delta} = (1-w)(yX_t - c_{t-\Delta}).$$

From this equation and continuity of $c$, we can now obtain the partial derivatives of $c$ with respect to $t$ by fixing $X_t = x$ and taking the limit $\Delta \to 0$. Similarly, the partial derivatives of $c$ with respect to $x$ are obtained directly from fixing $X_t = x$ in (24). The derivatives are given by

$$\frac{\partial}{\partial t}c_t = (1-w)(yx - c_t),$$

$$\frac{\partial}{\partial x}c_t = (1-w)y,$$

$$\frac{\partial^2}{\partial x^2}c_t = 0.$$

Based on the dynamics of $c$ from Itô's formula and the derivative specified above, we obtain the continuous-time dynamics of consumption under the hybrid strategy,

$$
\begin{aligned}
dc_t &= (1-w)(yX_t - c_t)dt + (1-w)y dX_t \\
&= \left(\varphi(1 + r + \mu_t^* + \pi_t(\lambda - r))X_t - (1-w)(1+y)c_t\right)dt \\
&\quad + \varphi\pi_t\sigma X_t dW_t, \\
c_0 &> 0,
\end{aligned}
\tag{25}
$$

where we insert the wealth dynamics given by (6) and define $\varphi := (1-w)y$ in the second equality (see Kashif et al. 2020). In principle, we could consider $w$ and $y$ as time-dependent since we consider continuous time in (25). But we will keep $w$ and $y$ as constants in this section.

Observe that the consumption dynamics can be rewritten as

$$
\begin{aligned}
dc_t &= (1-w)(1+y)\left(\frac{y(1 + r + \mu_t^* + \pi_t(\lambda - r))X_t}{1+y} - c_t\right)dt \\
&\quad + \varphi\pi_t\sigma X_t dW_t, \\
c_0 &> 0,
\end{aligned}
\tag{26}
$$

Hence, consumption under the hybrid strategy is a mean reverting process where $(1 - w)(1 + y)$ is the strength of mean reversion. The process reverts toward

$$\frac{y(1 + r + \mu_t^* + \pi_t(\lambda - r))X_t}{1 + y}.$$

The mean reversion property results in a smoothing effect on consumption, as noticed and analyzed by Kashif et al. (2020). Hence, this is an underlying argument that the hybrid strategy is an ad hoc smoothing strategy for consumption.

Including consumption as an additional state variable, investment is the only decision variable in the utility maximization problem. We find the optimal investment portfolio by solving the value function given by

$$V(t, x, c; \pi) = \sup_{\pi} E_{t,x,c} \left[ \int_t^T e^{-\int_t^s (\rho + \mu_u) du} \frac{1}{1 - \gamma} X_T^{1-\gamma} \varepsilon_T(s) \right], \tag{27}$$

subject to the wealth dynamics (6) and the consumption dynamics (25). Here, $E_{t,x,c}$ is the conditional expectation given that the investor is alive at time $t$, holds wealth $X_t = x$, and consumes $c_t = c$. We omit the minimum subsistence level of wealth in the utility maximization problem in Kashif et al. (2020). Thus, the investor obtains utility solely from terminal wealth under the hybrid strategy. The corresponding HJB equation

$$0 = V_t - (\rho + \mu)V + \sup_{\pi} \Bigg\{ ((r + \mu^* + \pi(\lambda - r))x - c) V_x$$

$$+ (\varphi(1 + r + \mu^* + \pi(\lambda - r))x - (1 - w)(1 + y)c) V_c + \frac{1}{2}\pi^2\sigma^2 x^2 V_{xx}$$

$$+ \frac{1}{2}\varphi^2\pi^2\sigma^2 x^2 V_{cc} + \varphi\pi^2\sigma^2 x^2 V_{xc} \Bigg\}, \tag{28}$$

$$V(T, x, c) = \frac{1}{1 - \gamma}x^{1-\gamma}.$$

The solution to the HJB Equation (28) is given by the following value function:

$$V(t, x, c) = \frac{1}{1 - \gamma}a_3(t)^\gamma (x - b_3(t, c))^{1-\gamma}, \tag{29}$$

where

$$a_3(t) = e^{-\int_t^T \left( \tilde{r} + \tilde{\mu}_u - (1 - \frac{1}{\gamma})(1 + r + \mu_u^*)\varphi\eta(u) \right) du},$$

$$b_3(t, c) = c\eta(t).$$

Here, $\eta(t)$ is the solution to the Riccati equation given by

$$\dot{\eta}(t) = -(1 + r + \mu_t^*)\varphi\eta(t)^2 + (1 + r + \mu_t^* + \varphi - w)\eta(t) - 1,$$

$$\eta(T) = 0, \tag{30}$$

which can be rewritten as a second-order differential equation and solved numerically. The optimal investment portfolio under the hybrid strategy is given by

$$\pi_t^* = \frac{\lambda - r}{\sigma^2\gamma} \frac{X_t - c_t\eta(t)}{(1 - \varphi\eta(t))X_t}. \tag{31}$$

The derivation of the results follows partially Kashif et al. (2020) with $\mu_t = 0$ and $\mu_t^* = 0$ for all $t \geq 0$. The solution of the function $\eta$ differs, since we think of $\eta$ as a function of time with the terminal condition $\eta(T) = 0$, whereas Kashif et al. (2020) think of $\eta$ as a constant.

In line with the habit strategy, optimal investment under the hybrid strategy is time-dependent. Now, the actual investment, $\pi_t^* X_t$, is a proportion of the current wealth over a proportion of the current consumption level. The hybrid strategy reduces risk tolerance to maintain consumption at some non-guaranteed level. However, the optimal investment portfolio increases over time since the denominator of (31) decreases faster than the numerator. We found the optimal investment portfolio by maximizing the expected utility of terminal wealth. Thus, we increase the quantity invested in the risky asset to secure enough risk exposure to maximize the expected utility of terminal wealth.

It is beyond the scope of this paper to solve $\eta$ as the Riccati equation given by (30). Alternatively, we present a special case where the derivation of the solution to $\eta$ is more manageable, as stated in the following remark. It is attained by considering a specific construction of $\varphi$.

**Remark 1.** *If we consider $w$ and $y$ to be time-dependent and assume that*

$$\varphi(t) = (1 - w(t))y(t) = \frac{k_\eta}{\eta(t)},$$

*where $0 < k_\eta < 1$ is a constant, then the Riccati equation given by (30) becomes an ODE on the form*

$$\dot{\eta}(t) = \big((1 + r + \mu_t^*)(1 - k_\eta) - w(t)\big)\eta(t) - (1 - k_\eta),$$
$$\eta(T) = 0,$$

*which has a unique solution given by*

$$\eta(t) = (1 - k_\eta)\int_t^T e^{-\int_t^s ((1 + r + \mu_u^*)(1 - k_\eta) - w(u))du} ds.$$

*In this case, the function $\eta$ is equal to a constant $(1 - k_\eta)$ multiplied by an annuity determined by $(1 + r + \mu_t^*)(1 - k_\eta) - w(t)$.*

## 4. The Development of Consumption over Time

The dynamics of a process reveal the development of the process over time. This section aims to study the effect of optimal consumption under the three strategies. We find the optimal consumption dynamics in the classical and habit strategies by Itô's formula. In the hybrid strategy, we induce the already seen consumption dynamics by the optimal investment to find the optimal version of the dynamics.

### 4.1. Consumption Dynamics in the Classical Strategy

Applying Itô's formula to the optimal consumption in the classical strategy given by (14), the dynamics of optimal consumption are derived by

$$dc_t^* = \frac{\partial}{\partial t}c_t^* dt + \frac{\partial}{\partial x}c_t^* dX_t + \frac{1}{2}\frac{\partial^2}{\partial x^2}c_t^* (dX_t)^2. \tag{32}$$

Finding the derivatives of $c^*$, inserting these, the wealth dynamics with the optimal investment portfolio into (32) and rearranging, we can state the following result:

**Theorem 1.** *Consider optimal consumption in the classical strategy defined by (14). Then, the dynamics of optimal consumption are given by*

$$dc_t^* = \frac{r + \mu_t^* - \rho - \mu_t + \frac{1}{2\gamma}(1+\gamma)\theta^2}{\gamma} c_t^* dt + \frac{\theta}{\gamma} c_t^* dW_t, \tag{33}$$

$$c_0^* = \frac{x_0}{a_1(0)}, \tag{34}$$

*where the deterministic function $a_1$ is defined by* (11).

**Proof of Theorem 1.** See Appendix A.1. □

From Theorem 1, we note that optimal consumption in the classical strategy is a geometric Brownian motion. As Bruhn and Steffensen (2013), we deduce that the stock market fluctuations immediately affect the current consumption level due to the stochastic *dW*-term.

The *dt*-term reveals the expected development of consumption over time since a Brownian motion has a mean zero. Disregarding the risk aversion parameter in the denominator of the *dt*-term, we see that consumption increases by the interest rate, the pricing intensity, and the quantity obtained from investing in the risky asset, i.e., capital gains. Moreover, consumption decreases with the impatience factor and the mortality rate. Consequently, ignoring capital gains, an increasing or decreasing tendency in the consumption level can be explained by the relation between the interest rate and the impatience factor or the pricing intensity and the mortality rate.

Regarding the impact of the risk aversion parameter $\gamma$ on the development of consumption over time, observe that the *dt*-term and the *dW*-term explode as $\gamma \to 0$. On the other hand, both terms vanish as $\gamma \to \infty$. Hence, the risk-tolerant investor experiences a highly volatile development in consumption, and the risk-averse investor prefers a less volatile or constant consumption level. Thus, the risk-averse investor is averse to variation over time.

*4.2. Consumption Dynamics in the Habit Strategy*

Again, by applying Itô's formula to the optimal consumption in the habit strategy given by (21), the dynamics of optimal consumption are in the following form:

$$dc_t^* = \frac{\partial}{\partial t} c_t^* dt + \frac{\partial}{\partial x} c_t^* dX_t + \frac{\partial}{\partial h} c_t^* dh_t + \frac{1}{2} \frac{\partial^2}{\partial x^2} c_t^* (dX_t)^2. \tag{35}$$

Finding the partial derivatives of $c^*$, inserting these, the wealth dynamics, and the habit dynamics with the optimal investment portfolio into (35) and rearranging, we obtain the following result:

**Theorem 2.** *Consider optimal consumption in the habit strategy defined by* (21). *Then, the dynamics of optimal consumption are given by*

$$\begin{aligned}
dc_t^* = &\left[ \left( \frac{r + \mu_t^* - \rho - \mu_t + \frac{1}{2\gamma}(1+\gamma)\theta^2 + \kappa(t)}{\gamma} + \alpha \right) c_t^* \right. \\
&\left. - \left( \frac{r + \mu_t^* - \rho - \mu_t + \frac{1}{2\gamma}(1+\gamma)\theta^2 + \kappa(t)}{\gamma} + \beta \right) h_t \right] dt \\
&+ \frac{\theta}{\gamma}(c_t^* - h_t) dW_t,
\end{aligned} \tag{36}$$

$$c_0^* = h_0 + (1 + \alpha b_2(0))^{-\frac{1}{\gamma}} \frac{x_0 - h_0 b_2(0)}{a_2(0)},$$

*where*

$$\kappa(t) = \alpha \left( 1 - \frac{(r + \mu_t^* + \beta)b_2(t)}{1 + \alpha b_2(t)} \right). \tag{37}$$

*The deterministic functions $a_2$ and $b_2$ are defined by (19) and (20), respectively.*

**Proof of Theorem 2.** See Appendix A.2. □

From Theorem 2, we infer that optimal consumption under the habit strategy is less volatile than optimal consumption under the classical strategy. Now, stock market fluctuations immediately affect only consumption over the current habit level. Optimal consumption over the current habit level is a geometric Brownian motion, which can be obtained by a minor rewriting of (36) (see Bruhn and Steffensen 2013). This supports the idea of using the habit strategy as a smoothing strategy for consumption.

Furthermore, the incorporation of additive habit formation impacts the expected development of consumption. Now, consumption evolves as a proportion of the current consumption level over a proportion of the existing habit level. The proportions are identical to the fraction in the $dt$-term of optimal consumption in the classical strategy, including a quantity, $\kappa(t)$, added to the numerator as a result of introducing habit formation. Additionally, the habit preferences are added to the fraction.

Moreover, note that for $\alpha = \beta = 0$ and $h_t = 0$ for all $t \geq 0$ in Theorem 2, we arrive at the optimal consumption dynamics from Theorem 1. Hence, the structure of the dynamics allows us to switch between the classical and the habit strategy.

For another special case, observe that for $\gamma \to \infty$, the consumption dynamics in the habit strategy are equal to the habit dynamics given by (17). Thus, the optimal consumption of a risk-averse investor with additive habit formation in the preferences evolves as the habit level.

Next, recall the expression of optimal consumption under the habit strategy given by (21). Observe that $c^*$ is affine in $h$. Thus, isolating $h$ in the expression of $c^*$, we obtain an expression of the habit level in terms of optimal consumption and wealth:

$$h_t = \frac{a_2(t)c_t^* - (1 + \alpha b_2(t))^{-\frac{1}{\gamma}} X_t}{a_2(t) - b_2(t)(1 + \alpha b_2(t))^{-\frac{1}{\gamma}}}. \tag{38}$$

Inserting (38) into the dynamics of $c^*$ given by Theorem 2 and rearranging, we rephrase the optimal consumption dynamics as follows.

**Theorem 3.** *Consider the habit level given by (38). Then, the dynamics of optimal consumption in the habit strategy from Theorem 2 can be rewritten as*

$$
\begin{aligned}
dc_t^* &= f_1(t)c_t^* dt + f_2(t)X_t dt + g_1(t)c_t^* dW_t + g_2(t)X_t dW_t, \\
c_t^* &= 0,
\end{aligned} \tag{39}
$$

*where $f_1$, $f_2$, $g_1$ and $g_2$ are deterministic functions given by*

$$f_1(t) = \frac{(\alpha - \beta)a_2(t) - \left(\frac{r + \mu_t^* - \rho - \mu + \frac{1}{2\gamma}(1+\gamma)\theta^2 + \kappa(t)}{\gamma} + \alpha\right)b_2(t)(1 + \alpha b_2(t))^{-\frac{1}{\gamma}}}{a_2(t) - b_2(t)(1 + \alpha b_2(t))^{-\frac{1}{\gamma}}}, \quad (40)$$

$$f_2(t) = \frac{\left(\frac{r + \mu_t^* - \rho - \mu + \frac{1}{2\gamma}(1+\gamma)\theta^2 + \kappa(t)}{\gamma} + \beta\right)(1 + \alpha b_2(t))^{-\frac{1}{\gamma}}}{a_2(t) - b_2(t)(1 + \alpha b_2(t))^{-\frac{1}{\gamma}}}, \quad (41)$$

$$g_1(t) = -\frac{\theta}{\gamma}\frac{b_2(t)(1 + \alpha b_2(t))^{-\frac{1}{\gamma}}}{a_2(t) - b_2(t)(1 + \alpha b_2(t))^{-\frac{1}{\gamma}}}, \quad (42)$$

$$g_2(t) = \frac{\theta}{\gamma}\frac{(1 + \alpha b_2(t))^{-\frac{1}{\gamma}}}{a_2(t) - b_2(t)(1 + \alpha b_2(t))^{-\frac{1}{\gamma}}}. \quad (43)$$

**Proof of Theorem 3.** The result of Theorem 3 follows directly from inserting the habit level given by (38) into the optimal consumption dynamics given by (36) and gathering the *dt*-terms and *dW*-terms multiplied by $c^*$ and $X$, respectively. □

Hence, we have rephrased the dynamics of optimal consumption in the habit strategy without the habit level. This is reasonable when considering the design of a pension product. The visible variables in a pension product should solely be consumption and wealth, but the habit formation of the investor could reasonably drive the underlying mechanisms. But it is difficult as a pension company to form the value of the initial habit level of the investor, and now we have avoided this difficulty.

In addition, recall that the optimal investment under the habit strategy is given by

$$\pi_t^* = \frac{\lambda - r}{\sigma^2\gamma}\frac{X_t - h_t b_2(t)}{X_t}.$$

Inserting the habit level expressed by (38) into $\pi^*$, we obtain an expression of optimal investment without the habit level:

$$\pi_t^* = \frac{\lambda - r}{\sigma^2\gamma}\frac{a_2(t)(X_t - c_t^* b_2(t))}{\left(a_2(t) - b_2(t)(1 + \alpha b_2(t))^{-\frac{1}{\gamma}}\right)X_t}. \quad (44)$$

Hence, the optimal investment under the habit strategy has the same structure as the optimal investment in the hybrid strategy. Now, the actual investment, $\pi_t^* X_t$, is a proportion of the current wealth over a proportion of the current consumption level.

*4.3. Consumption Dynamics in the Hybrid Strategy*

Recall the dynamics of consumption in the hybrid strategy:

$$\begin{aligned} dc_t &= \left(\varphi(1 + r + \mu_t^* + \pi_t(\lambda - r))X_t - (1 - w)(1 + y)c_t\right)dt \\ &\quad + \varphi\pi_t\sigma X_t dW_t, \\ c_0 &> 0, \end{aligned} \quad (45)$$

with $\varphi = (1 - w)y$. We found that optimal investment under the hybrid strategy is given by

$$\pi_t^* = \frac{\lambda - r}{\sigma^2\gamma}\frac{X_t - c_t\eta(t)}{(1 - \varphi\eta(t))X_t}, \quad (46)$$

where $\eta$ solves the Riccati equation given by (30). Inserting (46) into the dynamics of *c* given by (45) and doing a minor rewriting, we obtain the following result:

**Theorem 4.** *Consider consumption in the hybrid strategy defined by* (24) *following the dynamics given by* (45). *With the optimal investment strategy given by* (46), *the consumption dynamics in the hybrid strategy can be rewritten as*

$$
dc_t = \tilde{f}_1(t)c_t dt + \tilde{f}_2(t)X_t dt + \tilde{g}_1(t)c_t dW_t + \tilde{g}_2(t)X_t dW_t,
$$
$$
c_0 > 0,
$$
(47)

*where* $\tilde{f}_1$, $\tilde{f}_2$, $\tilde{g}_1$ *and* $\tilde{g}_2$ *are deterministic functions given by*

$$
\tilde{f}_1(t) = -\frac{\theta^2}{\gamma}\frac{\varphi\eta(t)}{1 - \varphi\eta(t)} - (1-w)(1+y),
$$
(48)

$$
\tilde{f}_2(t) = \frac{\theta^2}{\gamma}\frac{\varphi}{1 - \varphi\eta(t)} + \varphi(1 + r + \mu_t^*),
$$
(49)

$$
\tilde{g}_1(t) = -\frac{\theta}{\gamma}\frac{\varphi\eta(t)}{1 - \varphi\eta(t)},
$$
(50)

$$
\tilde{g}_2(t) = \frac{\theta}{\gamma}\frac{\varphi}{1 - \varphi\eta(t)}.
$$
(51)

**Proof of Theorem 4.** The result of Theorem 4 follows directly from inserting the optimal investment strategy given by (46) into the consumption dynamics given by (45) and gathering the *dt*-terms and *dW*-terms multiplied by *c* and *X*, respectively. □

Note the astonishing similarities between the structure of the consumption dynamics from Theorems 3 and 4. We will elaborate and employ these similarities in the design of a smooth pension product in Section 5. We formalize the comparison of the two strategies in the following subsection.

*4.4. Comparing Consumption Dynamics in the Habit and the Hybrid Strategy*

In the preceding section, we compare the habit and hybrid strategies by comparing their consumption dynamics.

**Theorem 5.** *The consumption dynamics in the habit strategy given by Theorem 3 and the consumption dynamics in the hybrid strategy given by Theorem 4 coincide if*

$$
f_1(t) = \tilde{f}_1(t),
$$
(52)
$$
f_2(t) = \tilde{f}_2(t),
$$
(53)
$$
g_1(t) = \tilde{g}_1(t),
$$
(54)
$$
g_2(t) = \tilde{g}_2(t).
$$
(55)

**Proof of Theorem 5.** The result of Theorem 5 follows directly by comparing the consumption dynamics from Theorems 3 and 4. □

Hence, the habit strategy and the hybrid strategy for consumption are identical if Equations (52)–(55) in Theorem 5 are fulfilled. Even though the equations are not fulfilled, we still obtain that the structure of consumption and wealth is the same in the consumption dynamics under the two strategies. Thus, consumption under the two strategies evolves similarly but with (possibly) different deterministic functions. This remarkable result shows that the hybrid strategy complies with the preferences of additive habit formation. Hence, we conclude that the habit and hybrid strategies satisfy the same preferences.

## 5. Designing a Smooth Pension Product

This section aims to give two approaches for designing a smooth pension product, i.e., a smooth life annuity, which complies with the preferences of an investor who wants stability with respect to consumption and has no utility from their bequest. This is accomplished by inspiration of the habit and hybrid strategies and their similarities studied in Section 4. In addition, we incorporate optimal consumption under the classical strategy. We aim for a feasible, transparent, and fair design from the perspective of both the investor and the pension company. In designing a pension product, we need to specify the strategy for consumption and investment in the risky asset. With inspiration from the habit strategy and the hybrid strategy, this boils down to defining the constant and the weight in the discrete-time formulation of consumption under the hybrid strategy given by (24) and revisiting the optimal investment portfolio under the two strategies.

Consider the discrete-time formulation of consumption under the hybrid strategy given by (24), but now let $w$ and $y$ be time-dependent. Thus, we redefine consumption by

$$c_t = w(t)c_{t-1} + (1 - w(t))y(t)X_t. \tag{56}$$

In the design of a pension product, it is reasonable to let $w$ and $y$ be time-dependent. For feasibility and transparency, the liabilities of the pension company should be equal to the investor's wealth. For fairness, the investor's wealth should tend to zero as we reach termination, since we are considering a life annuity and assuming that the investor has no utility from the bequest. We can incorporate these criteria by allowing time-dependent $w$ and $y$.

Let $y$ be defined as follows

$$y(t) = \frac{1}{a_1(t)}, \tag{57}$$

where $a_1$ is given by (11) from the classical strategy. Recall that $a_1$ is interpreted as an annuity with the utility-adjusted interest rate as the calculation rate and the utility-adjusted mortality rate as the calculation of mortality. Thus, with $y$ defined as (57), we have that $y(t)X_t$ is the optimal consumption under the classical strategy, which we interpreted as a standard market rate life annuity.

Now, let $w$ be defined as follows

$$w(t) = \begin{cases} \frac{\alpha a_w(t)}{1 + \alpha a_w(t)}, & \text{for } t \in (0, T], \\ 0, & \text{for } t = 0, \end{cases} \tag{58}$$

where $\alpha$ is a non-negative constant, and we define $a_w$ as an annuity given by

$$a_w(t) = \int_t^T e^{-\int_t^s (r_w + \tilde{\mu}_u)dy} ds, \tag{59}$$

with the calculation rate given by an interest rate, $r_w$, and the utility-adjusted mortality rate as the calculation of mortality.

The reasoning behind the construction of $w$ given by (58) is that the impact of last year's consumption level should depend on where the investor is in the decumulation phase. At the beginning of the decumulation phase, we want a higher weight on last year's consumption level to stabilize consumption. At the end of the decumulation phase, we want a higher weight on the future, i.e., a lower weight on past consumption, to let the standard market rate life annuity take over and the wealth go to zero. Observe that this reasoning is achieved by $w$ defined as (58) since $\alpha a_w(t)/(1 + \alpha a_w(t)) \to 0$ for $t \to T$. We use $a_w$ instead of $a_1$ since we want to change the interest rate to adjust the value of $w$. The higher the value of $r_w$, the lower the value of $w$. Inspired by the definition of the

habit level given by (16), we weigh the annuity by $\alpha$ to provide relative importance to past consumption.

Requiring that $w(t) = 0$ for $t = 0$, we set the initial value of consumption equal to $x_0/a_1(0)$, where $x_0$ is the value of the wealth at the time of retirement. Thus, we assume that the initial consumption value in the smooth pension product is equal to that in the classical strategy. This is also reasonable when comparing the two strategies.

Summarizing the above assumptions of $w$ and $y$, we rewrite consumption given by (56) as

$$
\begin{aligned}
c_t &= \frac{\alpha a_w(t)}{1 + \alpha a_w(t)} c_{t-1} + \frac{1}{1 + \alpha a_w(t)} \frac{X_t}{a_1(t)}, \\
c_0 &= \frac{x_0}{a_1(t)}.
\end{aligned}
\tag{60}
$$

Hence, we have specified the strategy for consumption in the smooth pension product as a hybrid between a standard market rate life annuity and stability with respect to last year's consumption level.

Regarding the investment portfolio in the smooth pension product, we need to reconsider how the optimal investment portfolio in the hybrid strategy was found before using the result given by (31) from this strategy. The optimal investment portfolio was found by optimizing the expected utility of terminal wealth similar to Kashif et al. (2020). This is a somewhat artificial construction when designing a smooth pension product where we want to let the wealth go to zero as we reach termination. However, by Remark 1, we observe a simple structure of the optimal investment portfolio under the hybrid strategy. We wish to examine how this investment portfolio works with consumption specified as (60). This constitutes the first approach to a smooth pension product.

To obtain even more feasibility and transparency of the investment portfolio in the smooth pension product, we become inspired by the optimal investment portfolio under both the habit and hybrid strategies. We employ the structure of the optimal investment portfolio under the habit strategy. Instead of subtracting a proportion of the current habit level from the wealth, we subtract a proportion of the current consumption level as in the optimal investment portfolio under the hybrid strategy given by (22). Moreover, we avoid a term multiplied by the wealth in the denominator as in the optimal investment portfolio under the habit strategy. This constitutes the second approach to a smooth pension product.

Hence, we have two approaches for the investment portfolio in the smooth pension product. With $y$ and $w$ given by (57) and (58), respectively, such that consumption is given by (60), we formalize two approaches for the smooth pension product.

*5.1. First Approach*

In the first approach, we employ the simple structure obtained by Remark 1. From the remark, we have that

$$
\eta(t) = \frac{k_\eta}{\varphi(t)} = \frac{k_\eta}{(1 - w(t))y(t)} = \begin{cases} k_\eta a_1(t)(1 + a_w(t)), & \text{for } t \in (0, T], \\ k_\eta a_1(t), & \text{for } t = 0, \end{cases}
\tag{61}
$$

where we insert $y$ and $w$ defined by (57) and (58), respectively, in the last equality. Inserting (61) into the optimal investment portfolio under the hybrid strategy given by (31), we specify the investment portfolio in the first approach for the smooth pension product by

$$
\begin{aligned}
\pi_t &= \frac{\lambda - r}{\sigma^2 \gamma} \frac{X_t - k_\eta a_1(t)(1 + a_w(t))c_t}{(1 - k_\eta)X_t}, \\
\pi_0 &= \frac{\lambda - r}{\sigma^2 \gamma} \frac{x_0 - k_\eta a_1(0)c_0}{(1 - k_\eta)x_0}.
\end{aligned}
\tag{62}
$$

Thus, the investment portfolio's denominator equals a constant multiplied by the wealth. Moreover, the proportion subtracted from the wealth is the current consumption level multiplied by a function similar to an annuity.

### 5.2. Second Approach

In the second approach, we set the denominator of the optimal investment portfolio under the hybrid strategy given by (31) equal to one multiplied by the wealth. Moreover, we specify the term subtracted from the wealth in the numerator by the current consumption level multiplied by an annuity. The construction is established by inspiration of the optimal investment portfolio under the habit strategy with the expression of the habit level in terms of optimal consumption and wealth inserted (see (44)). Thus, we define the investment portfolio in the second approach by

$$\pi_t = \frac{\lambda - r}{\sigma^2 \gamma} \frac{X_t - a_w(t)c_t}{X_t}, \tag{63}$$

for all $t \in [0, T]$, where the annuity $a_w$ is given by (59). Compared to the first approach, the second approach attempts to achieve even more simplicity and transparency in the structure of the investment portfolio.

In the first and second approaches, the term multiplied by the current consumption level tends to zero as we reach termination. Furthermore, note that if we choose a high value of the interest rate, $r_w$, in $a_w$, then the term becomes small. Thus, it is interpreted as the lack of protection against a decline in the market in the attempt to achieve stability concerning consumption.

### 5.3. Valuation

With the structure of consumption and investment portfolio in the two approaches for a smooth pension product given above, we have that the liabilities of the pension company are precisely equal to the investor's wealth. This holds by the fact that consumption, given by (60), tends to the optimal consumption under the classical strategy such that the whole wealth is paid out when we reach termination. Thus, we omit the complexity of setting aside a buffer to achieve a smoothing effect on consumption, which is the case in the existing smooth pension product Tidspension by Bruhn and Steffensen (2013). The simplicity of the two approaches also relies on the fact that the consumption level is non-guaranteed. Instead, we aim at a less volatile development of consumption.

We have obtained two feasible, transparent, and fair designs considering everything. Compared to the existing smooth pension products, we have increased the simplicity.

### 6. Numerical Examples

In this section, numerical examples of consumption over time illustrate the smoothing effect on consumption under the two approaches for a smooth pension product. We compare the development of wealth, consumption, and investment portfolios under the two approaches and the classical strategy. See Munk (2008) for a numerical study of optimal consumption and investment with additive habit formation in preferences. See Kashif et al. (2020) for numerical examples of the hybrid strategy.

The first subsection below establishes the numerical setup by specifying the parameters to simulate the simplified financial market. Additionally, we decide upon a representative estimation of the mortality rate and the pricing intensity. In the second subsection, we illustrate and analyze the development of wealth, optimal consumption, and investment portfolio in the classical strategy. Continuing to the third and the fourth subsection, we give numerical examples of the two approaches for a smooth pension product from Section 5. These examples are compared with the numerical results of the classical strategy from the second subsection.

### 6.1. Numerical Setup

The Euler scheme is used to simulate the development of the wealth process by a time-discretized approximation with independent normal random variables representing the Brownian motion for every year in the fixed time interval $[0, T]$. The simulation is repeated $n = 100{,}000$ times under the classical strategy and the two approaches, respectively, with the chosen values of the parameters from Table 1 and a specified model for the mortality rate and the pricing intensity as described below. We let the parameter $\alpha$ be equal to 0.2 aligned with Munk (2008). The rest of the parameters are chosen in line with comparable life insurance literature (see, e.g., Bruhn and Steffensen (2011) or Konicz et al. (2015)). The time frame used for the simulation under the classical strategy is 3.56 s, while it is 30.08 s under the first approach, and 24.66 s under the second approach.

**Table 1.** Parameter values needed for simulating the simplified financial market.

| Parameter | Descributon | Value |
|:---:|:---:|:---:|
| $x_0$ | Initial wealth | 100 |
| $T$ | Fixed time horizon | 45 |
| $\rho$ | Impatience factor | 0.04 |
| $\gamma$ | Risk aversion parameter | 2 |
| $r$ | Drift of the risk-free asset | 0.02 |
| $\lambda$ | Drift of the risky asset | 0.05 |
| $\sigma$ | Volatility of the risky asset | 0.2 |

As we only consider the decumulation phase of the investor, the fixed investment time interval $[0, T]$ represents that the investor retires at time 0 with age 65, and we assume the specified time horizon $T = 45$ at which the investor is age 110.

Regarding the lifetime uncertainty, we assume that $\mu_t = \mu_t^*$, i.e., zero market price of insurance risk, is modeled by a linear interpolation of the benchmark mortality rate for Danish women in 2021 over age 65 constituted by the Danish Financial Supervisory Authority (see DFSA 2022b). We omit to consider expected future lifetime improvements. Thus, we assume that a 65-year-old woman today in 2023 has the same mortality rate as a 65-year-old woman in 2021. Next year, when she turns 66, we think she will have the same mortality rate as a 66-year-old woman in 2021. A more realistic model for the mortality rate could have been constructed, but it is not the focus of this paper.

All numerical results are based on the assumption of no utility from bequest such that full mortality credits are assigned to the investor's wealth until death. The case of full utility from bequest, in the sense that no insurance is wanted, can, essentially, be obtained by setting $\mu_t = \mu_t^* = 0$ in all the formulas above. From a financial point of view, it corresponds to lowering the return rates $r$ and $\alpha$ simultaneously by $\mu_t$. This would lead to lower levels of consumption but along the same curvature as below. It is beyond the scope of this paper to examine in full the impact of utility from bequest, though. A different interesting case is where there is no utility from bequest but also no access to a life insurance market. The welfare loss from losing access to the insurance market is the main objective of Steffensen and Søe (2023) where there is no attention to smoothing, though.

### 6.2. The Classical Strategy

With the numerical setup from the preceding Section 6.1, we demonstrate the development of wealth, optimal consumption, and investment portfolio under the classical strategy. The optimal consumption and investment portfolio are given by (14) and (15), respectively. The expected development over time, accompanied by the 5% and 95% quantile, illustrates the strategy's volatility.

We leave out the graph of the optimal investment portfolio in the classical strategy as it is constant over time. With parameter values from Table 1, the optimal quantity invested in the risky asset is

$$\pi_t^* = \frac{\lambda - r}{\sigma^2 \gamma} = 0.375,$$

for all $t \in [0, T]$. Thus, about one-third of the wealth is invested in the risky asset throughout the entire decumulation phase.

Figure 1 shows that the investor is more eager to consume at the beginning of the decumulation phase than at the end, as we observe a decreasing graph of expected optimal consumption. An explanation of this development can be found by inspecting the drift term in the dynamics of optimal consumption under the classical strategy given in Theorem 1. Disregarding the return of the risky asset, the drift term contains the interest rate subtracted from the impatience factor, and as the values of these parameters are chosen such that $\rho > r$, we experience a decline in consumption over time. Hence, the investor is impatient.

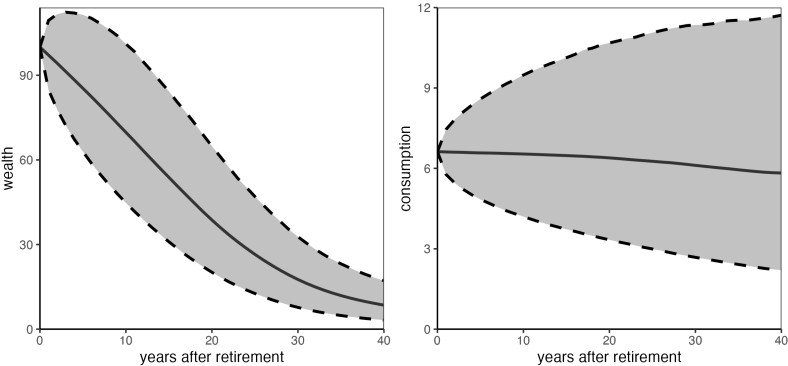

**Figure 1.** Wealth and consumption under the classical strategy simulated with parameter values from Table 1. The solid lines show the mean of 100,000 simulations, and the dashed lines show the 5% and 95% quantile.

Moreover, the wealth tends to zero as we reach expiry; i.e., the entire wealth is consumed since the investor has no utility from the bequest. This observation aligns with the insurance company's liabilities, which are repealed at expiry or the investor's death. Hence, the figure shows that the liabilities are equal to the wealth.

The 5% and 95% quantile emphasized by the dashed lines in Figure 1 visualize that 90% of the simulated paths of wealth and optimal consumption lie within this interval indicated by the shaded area. The path volatility expresses how the risky asset investment affects wealth development and optimal consumption over time. As observed by the dynamics of optimal consumption under the classical strategy from 1, consumption is immediately influenced by the volatility in the financial market. This uncertainty might be undesirable to an investor who favors stability in consumption over time. Hence, the optimal investment portfolio in the classical strategy is too risky to comply with the preferences of this type of investor.

*6.3. The First Approach of a Smooth Pension Product*

Now, we illustrate the first approach of a smooth pension product, where consumption and investment portfolio are specified as (60) and (62), respectively. The developments of wealth and consumption are shown in Figure 2. We examine the sensitivity of wealth and consumption to different values of the interest rate $r_w$ in the annuity $a_w$.

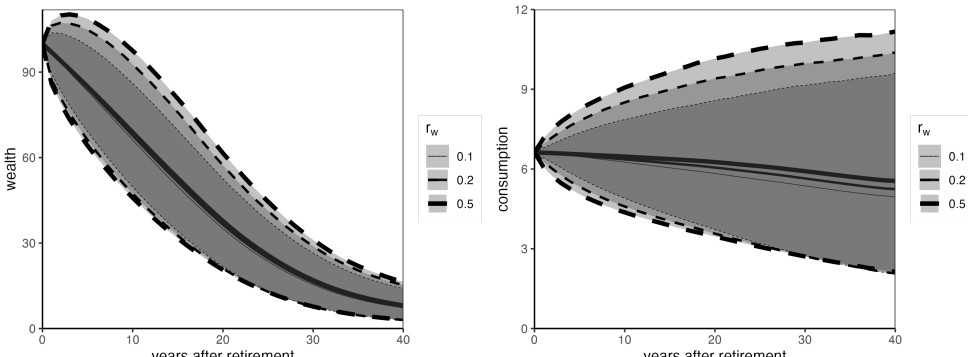

**Figure 2.** Wealth and consumption under the first approach of a smooth pension product simulated with different values of the interest rate $r_w$, $\alpha = k_\eta = 0.2$ and parameter values from Table 1. The solid lines show the mean of 100,000 simulations, and the dashed lines show the 5% and 95% quantile.

By Figure 2, we observe that the development of consumption under the first approach is less volatile than consumption under the classical strategy. The lower the value of $r_w$, the less volatility of consumption. This holds since the lower the value of $r_w$, the higher the value of $w$. Thus, for lower values of $r_w$, we maintain a higher proportion of last year's consumption level in calculating the current consumption level. At the same time, we observe that consumption under the first approach tends to consumption under the classical strategy as the value of $r_w$ increases. This holds since the weight, $w$, becomes small for higher values of $r_w$. By constructing consumption in the smooth pension product, consumption under the classical strategy takes over. Additionally, we observe a lower expected level of consumption for lower values of $r_w$. It is a compensation for maintaining a higher proportion of last year's consumption level.

As we approach termination, the volatility of consumption increases. An explanation of this can be found in developing the investment portfolio under the first approach.

Unlike the classical strategy, the investment portfolio under the first approach of a smooth pension product is time-dependent. Figure 3 visualizes the impact of different values of $r_w$ on the investment portfolio. The investment portfolio has different initial values for different values of $r_w$: a lower value of $r_w$ leads to a lower initial investment portfolio value. Thus, we invest less in risky assets to maintain a higher proportion of last year's consumption.

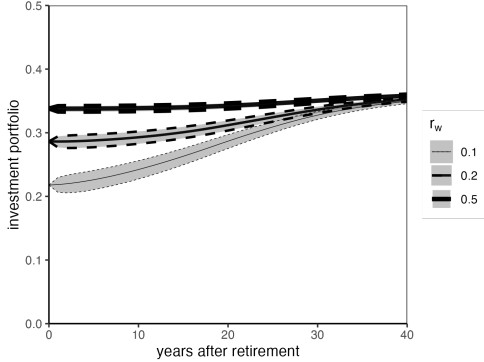

**Figure 3.** Investment portfolio under the first approach of a smooth pension product simulated with different values of the interest rate $r_w$, $\alpha = k_\eta = 0.2$ and parameter values from Table 1. The solid lines show the mean of 100,000 simulations, and the dashed lines show the 5% and 95% quantile.

Moreover, the investment portfolio is increasing and tending to the same value for the different values of $r_w$. The increasing development of the investment portfolio holds because the first approach is a special case of the hybrid strategy, where the optimal investment portfolio results from maximizing the expected utility of terminal wealth.

However, by constructing consumption under the smooth pension product, we let the wealth go to zero as we reach expiry, which we observe by Figure 2. The increasing investment portfolio is also why we keep increasing volatility in consumption over time.

At last, we substantiate the above analysis by illustrating the function $\eta$ given by (61) and the weights, $y$ and $w$, defined by (57) and (58), respectively, for different values of $r_w$.

By Figure 4, we only observe one graph of $y$, since $y$ is independent of $r_w$. Thus, the graphs of $y$ for different values of $r_w$ are indistinguishable. Note that the graph of $y$ is increasing over time and is rapidly closer to expiry such that the rest of the wealth is paid out by the construction of consumption under the smooth pension product.

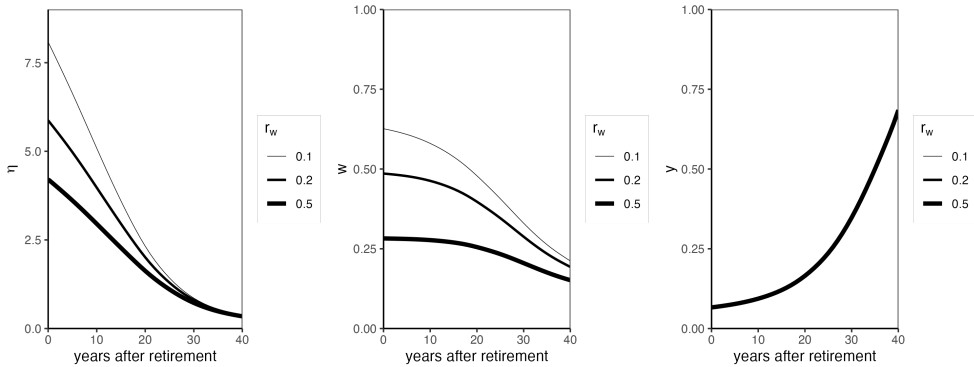

**Figure 4.** The function $\eta$ and the weights, $w$ and $y$, under the first approach of a smooth pension product with different values of the interest rate $r_w$, $\alpha = k_\eta = 0.2$ and parameter values from Table 1.

The graphs of $w$ in Figure 4 show that the weight decreases over time such that the standard market rate life annuity takes over as we reach expiry. For lower values of $r_w$, the higher the initial value of $w$ and vice versa. Thus, we maintain a higher proportion of last year's consumption level for lower values of $r_w$. Conversely, the standard market rate life annuity is more dominant for higher values of $r_w$.

Moreover, the function $\eta$ decreases rapidly through the entire decumulation phase. This contributes to the fact that we observe an increasing investment portfolio since the numerator decreases slower than the denominator by the structure of Equation (62). Additionally, the lower the value of $r_w$, the higher the value of $\eta$. Thus, for higher values of $r_w$, we subtract a higher proportion of the current consumption level to determine the quantity invested in the risky asset. This dampens the risk taking at the beginning of the decumulation phase, as Figure 3 shows.

### 6.4. The Second Approach of a Smooth Pension Product

Next, we illustrate the second approach of a smooth pension product, where consumption and investment portfolio are defined by (60) and (63), respectively. The development of wealth and consumption is shown in Figure 5. Again, we study the sensitivity of wealth and consumption to different values of the interest rate $r_w$ in the annuity $a_w$.

By Figure 5, we observe that the development of consumption under the second approach is less volatile than consumption under the classical strategy and consumption under the first approach. The same smoothing effect is reflected in wealth development under the second approach. Both wealth and consumption become less volatile for lower values of $r_w$. Note that consumption volatility increases at the beginning of the decumulation phase but stabilizes later in the decumulation phase. The stabilization of volatility happens sooner for lower values of $r_w$. This observation is supported by developing the investment portfolio under the second approach, as shown in Figure 6.

As in the first approach, the investment portfolio of a smooth pension product under the second approach is time-dependent. Figure 6 presents the impact of different values of $r_w$ on the investment portfolio. Again, we observe different initial investment portfolio values for different values of $r_w$. In contrast to the first approach, we observe that the

investment portfolio is decreasing over time, which explains the stabilization of volatility in Figure 5. As opposed to both the first approach and the classical strategy, the quantity invested in the risky asset is smaller compared to the three graphs of the expected investment portfolio with the respective corresponding value of $r_w$. Hence, to obtain less consumption volatility, the risk appetite is reduced.

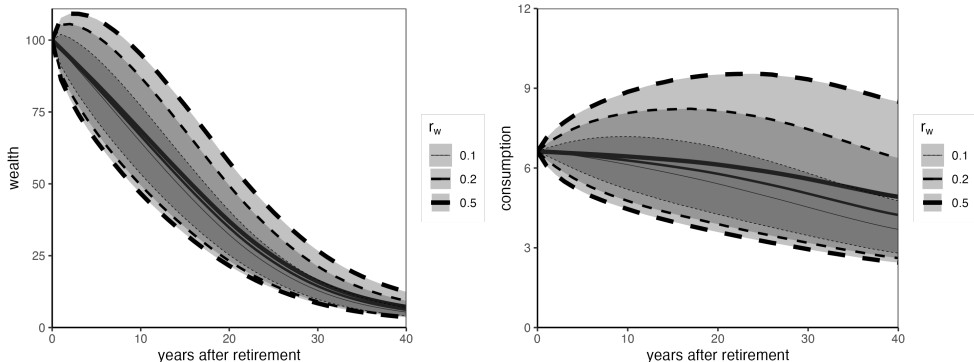

**Figure 5.** Wealth and consumption under the second approach of a smooth pension product simulated with different values of the interest rate $r_w$, $\alpha = 0.2$ and parameter values from Table 1. The solid lines show the mean of 100,000 simulations, and the dashed lines show the 5% and 95% quantile.

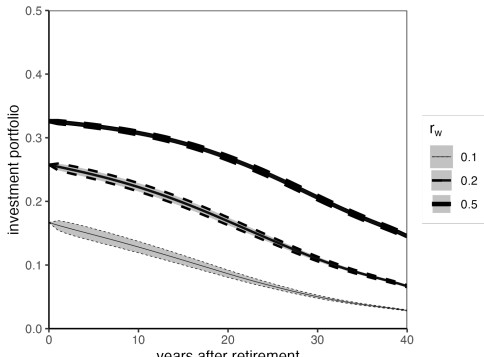

**Figure 6.** Investment portfolio under the second approach of a smooth pension product simulated with different values of the interest rate $r_w$, $\alpha = 0.2$ and parameter values from Table 1. The solid lines show the mean of 100,000 simulations, and the dashed lines show the 5% and 95% quantile.

Finally, we assist the above analysis by considering the annuity $a_w$ given by (59), which is multiplied by the current consumption level and subtracted from the wealth in the investment portfolio under the second approach given by (63). In Figure 7, we present $a_w$ and the weights, $y$ and $w$, defined by (57) and (58), respectively, for different values of $r_w$.

Note that the graphs of the weights in Figure 7 are identical to those in Figure 4. This holds since the definition of consumption and the weights are similar under the two approaches. Thus, regarding the weights, the same analysis and interpretation remain as above.

Figure 7 shows that the graphs of the annuity $a_w$ decrease over time. The higher the value of $r_w$, the higher the initial value of $a_w$. Thus, for lower values of $r_w$, we subtract a higher proportion, i.e., $a_w$, of the current consumption level directly from the actual amount invested in the risky asset by the construction of the investment portfolio in the second approach given by (63). As we subtract a proportion directly from the amount invested in the risky asset, the risk taking is dampened in the entire decumulation phase for all values of $r_w$ as observed by Figure 6. The value of $r_w$ dictates how much the risk taking is dampened.

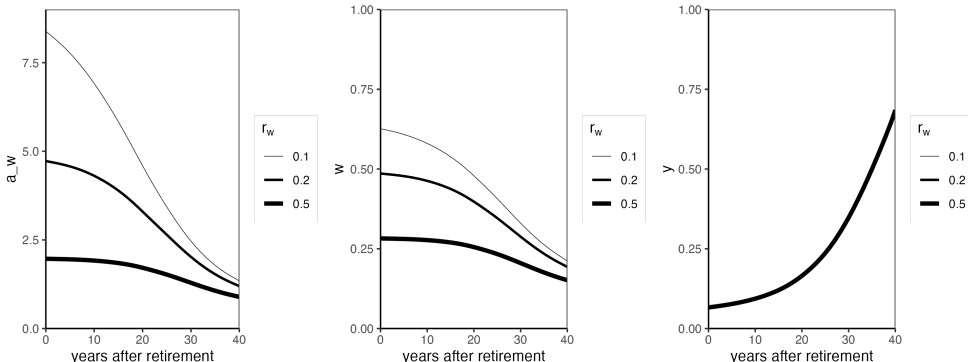

**Figure 7.** The annuity $a_w$ and the weights, $w$ and $y$, under the second approach of a smooth pension product with different values of the interest rate $r_w$, $\alpha = 0.2$ and parameter values from Table 1.

## 7. Conclusions

With inspiration from three strategies for consumption and investment, we have managed to design two approaches to a smooth pension product. The three strategies are the classical strategy, the habit strategy, and the hybrid strategy. Combining and modifying these strategies engenders the two approaches.

It turns out that optimal consumption under the classical strategy has the same structure as a standard market rate life annuity. Deriving the consumption dynamics under the classical and hybrid strategies, we observe that additive habit formation in preferences leads to the request for consumption stability. Comparing the consumption dynamics under the habit strategy and the hybrid strategy, we observe that the structures of the dynamics coincide. Hence, we deduce that the hybrid strategy meets the same preferences as the habit strategy. We wanted to understand these preferences to design a smooth pension product.

The objective of the smooth pension product is not to guarantee the same consumption level each year, but it is an attempt to smoothen the volatility of consumption. Moreover, we aim for a smooth pension product that is feasible, transparent, and fair from the perspective of both the investor and the pension company. Highly inspired by the consumption structure under the hybrid strategy, we let consumption be a weighted average of last year's consumption level and consumption under the classical strategy, i.e., a hybrid between stability with respect to last year's consumption level and a standard market rate life annuity. We let the weight be time-dependent such that the traditional market rate life annuity takes over as we reach termination. Hence, the pension company's liabilities equal the investor's wealth. Thus, we have obtained the smooth pension product's feasibility, transparency, and fairness. Reconsidering and inspired by both the hybrid strategy and the habit strategy, we give two approaches for the investment portfolio of the smooth pension product.

The development of the wealth process is simulated under the classical strategy and the two approaches, respectively. Each simulation is repeated $n = 100,000$, and the time frame under the classical strategy is 3.56 s, while it is 30.08 s under the first approach and 24.66 s under the second approach. The numerical examples show that consumption under the first and second approaches for the smooth pension product is less volatile than consumption under the classical strategy, i.e., a standard market rate life annuity. We see that consumption under the second approach is even less volatile than the first. An explanation for this result can be found in developing the investment portfolio under the respective approaches.

In the first approach, the quantity invested in the risky asset increases over time because the first approach is a special case of the hybrid strategy. The optimal investment portfolio under the hybrid strategy is found by maximizing the expected utility of terminal wealth. Thereby, we observe an increasing investment portfolio under the first approach.

The second approach is established to obtain an even simpler investment portfolio structure. It is inspired by the structure of the result obtained under the habit strategy with the expression of the habit level in terms of optimal consumption and wealth inserted.

For further research, we would study whether maximizing the expected utility of intertemporal consumption is possible given specified consumption dynamics when finding the optimal investment portfolio under the hybrid strategy. We claim that this changes the final result under the hybrid strategy.

We observed that the development of the investment portfolio impacts consumption volatility. Thus, we might have found the dynamics of the investment portfolio under the three strategies to increase the understanding of the development over time. Then, the influence of the different terms in the dynamics of the investment portfolio could be studied.

Finally, the investment portfolio under the second approach is low compared to the classical strategy. This is a result of smoothing the volatility of consumption. One might wonder whether the price of the smooth pension product is too high as the investor is dictated to take too low a risk. We need to avoid the smooth pension product becoming a guaranteed pension product. Thus, for further research, we could consider lower values of the risk aversion parameter $\gamma$ such that the investment portfolio under the second approach is increased at the beginning of the decumulation phase but still decreasing over time such that the volatility of consumption is still smoothed over time. Hence, we aim to take enough risk over the entire decumulation phase.

**Author Contributions:** Writing—original draft preparation, S.H.V.; writing—review and editing, M.S.; Supervision, M.S. All authors have read and agreed to the published version of the manuscript.

**Funding:** This research received no external funding.

**Institutional Review Board Statement:** Not applicable.

**Data Availability Statement:** See DFSA (2022b).

**Conflicts of Interest:** The authors declare no conflicts of interest.

## Appendix A. Studying Optimal Consumption

We give proof of the consumption dynamics under the classical and habit strategies in the first and the second subsections, respectively.

*Appendix A.1. Proof of Theorem 1—Consumption Dynamics in the Classical Strategy*

Applying Itô's formula to $c^*$ given by (14), we find the dynamics of optimal consumption in the classical strategy by

$$dc_t^* = \frac{\partial}{\partial t}c_t^* dt + \frac{\partial}{\partial x}c_t^* dX_t + \frac{1}{2}\frac{\partial^2}{\partial x^2}c_t^* (dX_t)^2. \tag{A1}$$

By fixing $X_t = x$, the derivatives of $c^*$ are

$$\frac{\partial}{\partial t}c_t^* = -\frac{x}{a_1(t)^2}\dot{a}_1(t),$$

$$\frac{\partial}{\partial x}c_t^* = \frac{1}{a_1(t)},$$

$$\frac{\partial^2}{\partial x^2}c_t^* = 0.$$

Inserting the derivatives of $c^*$ and the wealth dynamics induced by the optimal controls into (A1), we obtain that

$$dc_t^* = \left( -\frac{X_t}{a_1(t)^2}\dot{a}_1(t) + \frac{1}{a_1(t)}\left( (r + \mu_t^* + \pi_t^*(\lambda - r))X_t - c_t^* \right) \right) dt + \frac{1}{a_1(t)}\pi_t^* \sigma X_t dW_t.$$

Inserting the expression of $\pi^*$ given by (15), recognizing the expression of $c^*$ and rewriting, we obtain that

$$dc_t^* = \left( -\frac{1 + \dot{a}_1(t)}{a_1(t)} + r + \mu_t^* + \frac{\theta^2}{\gamma} \right) c_t^* dt + \frac{\theta}{\gamma} c_t^* dW_t. \tag{A2}$$

By Leibniz's integral rule, we have that

$$\dot{a}_1(t) = -1 + (\tilde{r} + \tilde{\mu}_t)a_1(t).$$

Thus, we see that

$$-\frac{1 + \dot{a}_1(t)}{a_1(t)} = -(\tilde{r} + \tilde{\mu}_t). \tag{A3}$$

Inserting (A3) together with the expression of $\tilde{r}$ and $\tilde{\mu}$ into (A2), then by a minor rewriting, we obtain the result of Theorem 1, where the initial condition follows directly by letting $t = 0$ and $X_0 = x_0$ in (14).

*Appendix A.2. Proof of Theorem 2—Consumption Dynamics in the Habit Strategy*

Applying Itô's formula to $c^*$ given by (21), we find the dynamics of optimal consumption in the habit strategy by

$$dc_t^* = \frac{\partial}{\partial t}c_t^* dt + \frac{\partial}{\partial x}c_t^* dX_t + \frac{\partial}{\partial h}c_t^* dh_t + \frac{1}{2}\frac{\partial^2}{\partial x^2}c_t^* (dX_t)^2. \tag{A4}$$

Fix $X_t = x$ and $h_t = h$. Then, by a combination of the product and the quotient rule, the derivative of $c^*$ with respect to $t$ is

$$\frac{\partial}{\partial t}c_t^* = -\frac{\alpha \dot{b}_2(t)(1 + \alpha b_2(t))^{-\frac{1}{\gamma}-1}}{\gamma}\frac{x - hb_2(t)}{a_2(t)}$$
$$+ (1 + \alpha b_2(t))^{-\frac{1}{\gamma}}\frac{-h\dot{b}_2(t)a_2(t) - (x - hb_2(t))\dot{a}_2(t)}{a_2(t)^2}.$$

By Leibniz's integral rule, we have that

$$\dot{a}_2(t) = -(1 + \alpha b_2(t))^{1-\frac{1}{\gamma}} + (\tilde{r} + \tilde{\mu}_t)a_2(t), \tag{A5}$$
$$\dot{b}_2(t) = -1 + (r + \mu_t^* + \beta - \alpha)b_2(t). \tag{A6}$$

Inserting the expression of $\dot{a}_2$, recognizing $c_t^* - h$ and gathering the terms multiplied by $c^*$ and those multiplied by $h$, we have that

$$\frac{\partial}{\partial t}c_t^* = \left( -\frac{\alpha \dot{b}_2(t)}{\gamma(1 + \alpha b_2(t))} + \frac{(1 + \alpha b_2(t))^{1-\frac{1}{\gamma}}}{a_2(t)} - \tilde{r} - \tilde{\mu}_t \right) c_t^*$$
$$- \left( -\frac{\alpha \dot{b}_2(t)}{\gamma(1 + \alpha b_2(t))} + \frac{\dot{b}_2(t)(1 + \alpha b_2(t))^{-\frac{1}{\gamma}}}{a_2(t)} + \frac{(1 + \alpha b_2(t))^{1-\frac{1}{\gamma}}}{a_2(t)} - \tilde{r} - \tilde{\mu}_t \right) h$$
$$=: C(t) - H(t).$$

The remaining necessary derivatives are

$$\frac{\partial}{\partial x} c_t^* = \frac{(1 + \alpha b_2(t))^{-\frac{1}{\gamma}}}{a_2(t)},$$

$$\frac{\partial}{\partial h} c_t^* = 1 - \frac{b_2(t)(1 + \alpha b_2(t))^{-\frac{1}{\gamma}}}{a_2(t)},$$

$$\frac{\partial^2}{\partial x^2} c_t^* = 0.$$

Inserting the derivatives of $c^*$, the wealth and the habit dynamics induced by the optimal controls into (A4), we obtain that

$$
\begin{aligned}
dc_t^* &= \left[ C(t) - H(t) + \frac{(1 + \alpha b_2(t))^{-\frac{1}{\gamma}}}{a_2(t)} \left( (r + \mu_t^* + \pi_t^*(\lambda - r))X_t - c_t^* \right) \right. \\
&\quad \left. + \left( 1 - \frac{b_2(t)(1 + \alpha b_2(t))^{-\frac{1}{\gamma}}}{a_2(t)} \right)(\alpha c_t^* - \beta h_t) \right] dt \\
&\quad + \frac{(1 + \alpha b_2(t))^{-\frac{1}{\gamma}}}{a_2(t)} \pi_t^* \sigma X_t dW_t \\
&= \left[ \left( -\frac{\alpha \dot{b}_2(t)}{\gamma(1 + \alpha b_2(t))} + \frac{(1 + \alpha b_2(t))^{1 - \frac{1}{\gamma}}}{a_2(t)} - \tilde{r} - \tilde{\mu}_t \right. \right. \\
&\quad \left. + \frac{\theta^2}{\gamma} - \frac{(1 + \alpha b_2(t))^{-\frac{1}{\gamma}}}{a_2(t)} + \alpha - \frac{\alpha b_2(t)(1 + \alpha b_2(t))^{-\frac{1}{\gamma}}}{a_2(t)} \right) c_t^* \\
&\quad - \left( -\frac{\alpha \dot{b}_2(t)}{\gamma(1 + \alpha b_2(t))} + \frac{\dot{b}_2(t)(1 + \alpha b_2(t))^{-\frac{1}{\gamma}}}{a_2(t)} + \frac{(1 + \alpha b_2(t))^{1 - \frac{1}{\gamma}}}{a_2(t)} \right. \\
&\quad \left. \left. - \tilde{r} - \tilde{\mu}_t + \frac{\theta^2}{\gamma} + \beta - \frac{\beta b_2(t)(1 + \alpha b_2(t))^{-\frac{1}{\gamma}}}{a_2(t)} \right) h_t \right] dt \\
&\quad + \frac{(r + \mu_t^*)X_t(1 + \alpha b_2(t))^{-\frac{1}{\gamma}}}{a_2(t)} dt + \frac{\theta}{\gamma}(c_t^* - h_t)dW_t. \tag{A7}
\end{aligned}
$$

In the second equation above, we insert the expression of $\pi^*$, recognize $c_t^* - h_t$, recall the definition of $C(t)$ and $H(t)$ and gather the $dt$ terms multiplied by $c^*$ and those multiplied by $h$.

Now, observe that three of the terms multiplied by $c^*$ vanish as

$$\frac{(1 + \alpha b_2(t))^{1 - \frac{1}{\gamma}}}{a_2(t)} - \frac{(1 + \alpha b_2(t))^{-\frac{1}{\gamma}}}{a_2(t)} - \frac{\alpha b_2(t)(1 + \alpha b_2(t))^{-\frac{1}{\gamma}}}{a_2(t)} = 0.$$

With the expression of $\dot{b}_2$ given by (A6), we see that

$$-\frac{\alpha \dot{b}_2(t)}{\gamma(1 + \alpha b_2(t))} = \frac{\alpha}{\gamma} \left( 1 - \frac{(r + \mu_t^* + \beta)b_2(t)}{1 + \alpha b_2(t)} \right).$$

Again with the expression of $\dot{b}_2$, two of the terms multiplied by $h$ read as

$$\frac{\dot{b}_2(t)(1+\alpha b_2(t))^{-\frac{1}{\gamma}}}{a_2(t)} - \frac{\beta b_2(t)(1+\alpha b_2(t))^{-\frac{1}{\gamma}}}{a_2(t)}$$

$$= \frac{(r+\mu_t^*)b_2(t)(1+\alpha b_2(t))^{-\frac{1}{\gamma}}}{a_2(t)} - \frac{(1+\alpha b_2(t))^{1-\frac{1}{\gamma}}}{a_2(t)}.$$

Remember that the two terms in the last equality above are multiplied by $h$. The last term cancels out with an identical term in the dynamics of $c^*$. The first term together with the $dt$ term in line (A7) are rewritten by

$$\frac{(r+\mu_t^*)X_t(1+\alpha b_2(t))^{-\frac{1}{\gamma}}}{a_2(t)} - \frac{(r+\mu_t^*)h_t b_2(t)(1+\alpha b_2(t))^{-\frac{1}{\gamma}}}{a_2(t)} = (r+\mu_t^*)(c_t^* - h_t).$$

Combining all the observations and using the expression of $\tilde{r}$ and $\tilde{\mu}$ to undergo a final rewriting, we obtain the dynamics of $c^*$ stated in Theorem 2. The initial condition follows directly by letting $t=0$ and $X_0 = x_0$ in (21).

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
