# Peer review of "On Smoothing and Habit Formation of Variable Life Annuity Benefits"

_jrfm, doi:10.3390/jrfm17020075_

Round 1
Reviewer 1 Report
Comments and Suggestions for Authors
-Title
In the title, the reviewer thinks it will be more relevant if the author mentions it in the Danish context. It is in line with the explanation in the background and the assumptions that the authors make in the simulation procedures.
- Introduction
Overall, the section has been clearly presented.
The authors have addressed why Denmark was chosen as the country to conduct the research. The aim of the research and the contribution of the study have been stated.
- Numerical example
For the numerical analysis, please give clear time frame data that the authors used for the simulation. The authors only mention 100.000 simulations without telling the timeframe. Please also disclose such information in the conclusion part.
- Others
Other than the above comments, the rest of the sections are fine.
Reviewer 2 Report
Comments and Suggestions for Authors
The paper is clear and well written. A few remarks:
* Authors indicate that the individual, in the even of not having utility of bequest, should receive the equivalent of mu* times the wealth as a counterpart from the insurer. They refer to Konincz et al. for justification. However, I do not see directly why this is needed. If the individual does not have a bequest motive, why should he get a premium from abstaining from bequest?
* Solutions to the HJB schemes. The authors in the classical and habit case state what the solution of the HJB is without giving more information on how this is obtained. For instance, how would you obtain (21)-(22) of value function. Which are the EDO and boundaries to satisfy?
* Equation (24): could the authors provide more details on the continuous time conterpart of (24) that is used to obtain the derivatives below line 237?
* In the numerical scheme the authors do not consider bequest. Hence, the wealth has to become zero by "maturity"/death. I am aware that it is beyond of the scope of the paper to add this further analysis, but it would be good to say something about how bequest might affect your results intuitively.
